# Living with rheumatic fever and rheumatic heart disease in Victoria, Australia: A qualitative study

Jane Oliver[1,2]*, Loudeen Fualautoalasi-Lam[1], Angeline Ferdinand[3], Ramona Tiatia[4], Bryn Jones[1], Daniel Engelman[1], Katherine B. Gibney[2‡], Andrew C. Steer[1‡]

**1** Murdoch Children's Research Institute, The Royal Children's Hospital, Parkville, Australia, **2** Department of Infectious Diseases, University of Melbourne, at the Peter Doherty Institute for Infection and Immunity, Melbourne, Australia, **3** Centre for Health Policy, Melbourne School of Population and Global Health, University of Melbourne, Victoria, Australia, **4** Department of Public Health, University of Otago Wellington, Wellington, New Zealand

‡ These authors are joint senior authors on this work.
* jane.oliver@unimelb.edu.au

**Data Availability Statement:** For participant privacy reason, the raw interview data is not available. Interview transcripts contain potentially identifying and highly sensitive patient information.

## Abstract

### Background

In Victoria, Australia, children with Pacific Islander ('Pacific') ethnicities are overrepresented in acute rheumatic fever (ARF) and rheumatic heart disease (RHD). In June 2023, ARF and RHD became notifiable in Victoria. To inform public health and clinical practice, we described young Pacific patients' and their caregivers' understandings and experiences of ARF/RHD, and identified possible ways to improve the delivery of clinical care.

### Methods

We established a project reference group including local Pacific people to guide this research. Pacific patients who attended an ARF/RHD clinic at The Royal Children's Hospital, Melbourne, were invited to participate, as were their caregivers. A Samoan researcher conducted qualitative 'talanoa' (conversational) interviews with patients and caregivers. A second researcher conducted semi-structured interviews with treating clinicians and other stakeholders. Interview transcripts underwent thematic analysis guided by the Tuilaepa Youth Mentoring Services Pacific Youth Wellbeing Framework.

### Results

We interviewed 27 participants. This included nine patients and nine caregivers, all of whom were Samoan. These 18 participants expressed a desire to learn more about ARF/RHD and connect with other affected people. While some shared their experiences of having well-liked and trusted healthcare providers, patients often struggled to have two-way clinical conversations. The need to support clinicians working with high-risk populations to improve their awareness of ARF was identified. Receiving treatment on time was a top priority for affected families, despite injection pain, inconvenience and financial costs. The need to support continuity of care for young adult patients was raised by participants.

Enquiries may be addressed to The Royal Children's Hospital Ethics Committee referencing HREC/91200/RCHM-2023 at rch.ethics@rch.org.au.

**Funding:** This work was funded by a Murdoch Children's Research Institute 2022 Stimulus Award (Early Career) made to JO (MCRIECA2022). Funding for this work was supplemented by a Leadership Investigator Grant from the National Health and Medical Research Council of Australia awarded to AS (APP #2009798). The funders had no role in the study design, data collection and analysis, decision to publish, or preparation of the manuscript. https://www.nhmrc.gov.au/.

**Competing interests:** The authors have declared that no competing interests exist.

## Conclusions

Pacific people living with ARF/RHD and their families require additional support to receive high quality management in Victoria. Introducing a patient register and a specialist RHD nurse would enhance access to treatment, as would removing cost barriers, improving clinical awareness of ARF/RHD and creating Victoria-specific patient resources.

## Author summary

Acute rheumatic fever (ARF) and its serious complication rheumatic heart disease (RHD) are key markers of social injustice. In Victoria, Australia, ARF and RHD became notifiable in mid-2022. This change was instigated in response to recent work showing an ongoing burden of disease in young Victorians, particularly people of Pacific Islander ('Pacific') ethnicities.

Until recently, nothing has been documented of the experiences that patients and their families have had when seeking clinical care for ARF/RHD in Victoria. We undertook this qualitative study to address this knowledge gap. Our study was co-designed by a project reference group which included local Pacific people. We used traditional Pacific 'talanoa' conversational style interviews to report on patients' and their caregivers' experiences of receiving clinical care.

Our findings provide critical guidance for best practice public health management in areas such as Victoria with a low but severely ethnically inequitable disease burden. We make recommendations to support clinicians in areas with at-risk populations to enhance their awareness of ARF and RHD, and improve the delivery of clinical care. Our recommendations provide direction for future initiatives to prevent and control these serious health conditions, and better support affected people.

## Introduction

Acute rheumatic fever (ARF) is an autoimmune response that occurs in a minority of cases following an untreated *Streptococcus pyogenes* infection. Once ARF occurs, regular antibiotic treatment ('secondary prophylaxis') is recommended to prevent recurrences [1]. ARF may lead to permanent cardiac damage termed 'rheumatic heart disease' (RHD). ARF typically presents in childhood, with incidence highest among those aged 5–14 years. RHD, which often presents in early-middle adulthood [1], is a major cause of preventable morbidity and early death in low-middle income countries. In 2015, 319,400 deaths were attributed to RHD globally, with the highest burden in Oceania, South Asia, and central sub-Saharan Africa [2]. ARF was once common throughout the world, but as living conditions, home crowding and access to antibiotics improved from the 1970s [3], ARF became recognised as a disease of social injustice[1,4]. Now rare in most high-income countries, ARF persists as a public health problem in low-middle income countries and some First Nations populations in high-income countries. Polynesian, Micronesian or Melanesian ('Pacific peoples'), Māori, and Aboriginal and Torres Strait Islander 'Indigenous' children experience among the highest ARF incidences in the world[1,4].

Australian clinical guidelines recommend secondary prophylaxis via deep intramuscular injection of benzathine penicillin G (BPG) every 21–28 days for at least five years following the initial ARF occurrence [5]. The World Health Organization recommends RHD control

programs be implemented in areas with a significant disease burden, with best-practice patient management facilitated though patient registers [6]. In Australia, secondary prophylaxis is currently coordinated by RHD registers in many, but not all, jurisdictions (the Northern Territory, Queensland, New South Wales (NSW), South Australia and Western Australia all have registers) [5,7]. The Australian ARF/RHD guidelines recognise that Pacific people who live in metropolitan households which are affected by crowding and/or lower socioeconomic status have a high risk of ARF [5]. In NSW, of the 87 cases added to the RHD register from October 2015 to December 2017, 37 (43%) were Pacific and 35 (40%) were Indigenous [8]. In 2016, <1% of the NSW population were Pacific and <3% were Indigenous [9]. In Victoria, ARF and RHD were made notifiable to the state Department of Health on 31 July 2023. There is no centralised RHD register in Victoria to provide a straightforward means of monitoring cases' BPG uptake. From 2010–2019, 108 Victorian resident children and adolescents were admitted to either of the state's two tertiary paediatric hospitals with ARF/RHD. These hospitals are Monash Children's Hospital and the Royal Children's Hospital, Melbourne (RCH) [10]. RCH is the major specialist paediatric hospital in Victoria. It is recommended that all suspected ARF/RHD cases in Victoria be referred there for assessment [11]. Of the 108 resident children, 49 (45%) were identified from hospital records as having Pacific ethnic backgrounds (including 36 Samoan children), and 11 (10%) were Indigenous. This equates to an annual incidence of 32 ARF/RHD cases per 100,000 Pacific children aged 5–14 years, and 4 per 100,000 Indigenous children in this age group, compared to an overall annual incidence of 0.8 per 100,000 children aged 5–14 years in Victoria [10].

As of the 2016 census, 62,613 Victorian residents indicated they had Pacific/Māori ancestry, of whom over 90% resided in metropolitan Melbourne [12]. Pacific people living in Victoria experience worse educational and employment outcomes than the Victorian average [12–14], with limited English language fluency reported among 20% of those aged >65 years and 3% of young adults [12].

The Fonofale Model of Health provides a definition of how and what Pacific people consider to be the major determinants of their health [15]. The Tuilaepa Youth Mentoring Services (TYMS) Pacific Youth Wellbeing Framework is a strengths-based adaption of the Fonofale Model specific to youth wellbeing [15,16]. Family is positioned as the foundation for health, and wellbeing includes body, mind, spirit and culture, taking into account broader contexts such as time and environment (Fig 1) [17].

Several reports have described Victorian Pacific communities as being 'forgotten' by government and mainstream society [18–20]. Reported barriers to healthcare include a tendency to avoid seeking conventional medical treatment unless seriously unwell [19,21], stigmatisation of illness [14,21,22], low health literacy and racism in healthcare settings [14,20,22]. A number of reports have considered unmet needs in the Victorian Pacific community, and strongly recommend removing barriers to welfare supports and education, including by expanding services available to Pacific youth [13,19,20,23].

A New Zealand analysis included 38 Māori and Pacific participants with ARF/RHD and identified barriers to accessing healthcare, including secondary prophylaxis. Key barriers were socioeconomic deprivation, racism in healthcare settings and age-inappropriate healthcare models. The authors identified particular service models and approaches taken by healthcare providers that helped improve BPG adherence and create positive experiences for families. The key facilitators for effective management of ARF they identified were: flexible, community-based models of care; good communication and rapport between patients, families and health professionals; and effective information-sharing and referral pathways between paediatric and adult health services, and between jurisdictions [24]. Despite being substantially over-represented in the Victorian burden of ARF/RHD, little has been documented of the patient journey for young Pacific people with these conditions in Victoria.

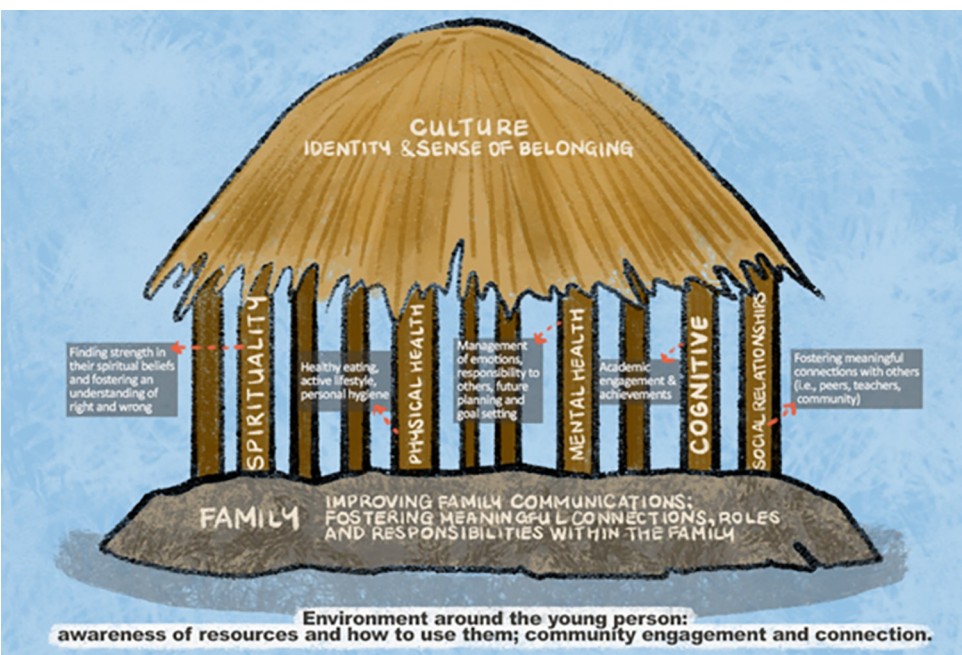

**Fig 1. The Tuilaepa Youth Mentoring Services Pacific Youth Wellbeing Framework [16].**

To inform public health and clinical practice in Victoria, we described young Pacific patients' and their caregivers' understandings and experiences of ARF/RHD, and identified possible ways to improve the delivery of clinical care.

## Methods

### Ethics

Ethics approval was provided by the Royal Children's Hospital Research Ethics Committee (#91200). All participants provided written informed consent. In addition, informed written consent was obtained from the parent/guardian for all participants aged less than 18 years.

This study is presented using the Consolidated Criteria for Reporting Qualitative Research (COREQ) [25]. A qualitative co-design methodology and a constructivist approach was used. This approach recognises how individuals' experiences are pivotal in shaping their knowledge and understandings of the world around them [26]. A Project Reference Group (PRG) guided all aspects of the research. The PRG was comprised of stakeholders including local and international Pacific community members, health professionals, and Victorian Government policy advisors. The PRG met twice, once prior to data collection to advise on study scope, aims and methodology, and once afterwards to help contextualise findings and advise on recommendations. The chairperson ensured that local Pacific community members' voices were given priority in meetings. Two of the researchers are Samoan (RT, LF) and the others are of New Zealand-European/Australian-European/Caribbean descent.

### Participant eligibility and recruitment

Patients with Pacific ethnic backgrounds who resided in Victoria and had attended an ARF/RHD outpatient clinic at The Royal Children's Hospital, Melbourne (RCH) since 1 June 2019 were identified through by a paediatrician who was associated with the clinics reviewing RCH

records. Their caregiver was eligible to participate in a qualitative interview, as was the patient if aged ≥10 years old. Patients aged ≥15 years old were also eligible to participate in a youth focus group. In addition, general practitioners (GPs) who treated eligible patients were invited to be interviewed, as were other stakeholders who worked with eligible ARF/RHD patients, or their data, with the aim of improving health outcomes.

The screening paediatrician phoned eligible patients aged ≥18 years or their caregiver if younger, briefly explained the study and requested permission to pass their contact information on to a researcher (LF). The researcher phoned those who agreed, discussed the study, obtained consent and later conducted interviews. Recruitment of patients and caregivers ceased when the researchers perceived thematic saturation had occurred as informed by the TYMS Pacific Youth Wellbeing Framework.

Eligible GPs were identified from RCH medical records. Their practices were contacted up to two times by email, and once by phone, by a second researcher (JO) who invited the GP to participate. Other stakeholders were contacted directly following their identification by the research team and/or PRG. Recruitment of GPs ceased once attempts to contact all eligible people had been made. Recruitment of other stakeholders ceased when the researchers perceived sufficient information to contextualise patient and caregiver interview findings had been gathered.

## Interviews

Interviews followed semi-structured interview guides (S1 and S3 Appendices) and were audio recorded. Interview questions were tailored to participants' roles and experiences. Interview topics addressed patient and caregiver understandings of ARF and RHD, and for clinicians/ other stakeholders, perceptions of how patients accessed and understood secondary prophylaxis. Interviews explored enablers and barriers to secondary prophylaxis, adherence and attending outpatient appointments. Potential ways to improve the delivery of clinical care were discussed, as were surveillance and health system gaps, particularly with stakeholders who worked in RHD prevention or control programs. Minor adaptations were made to the patient/caregiver interview guide following a review of the first two interview audios (by LF, JO, RT).

Patients and caregivers chose whether to be interviewed online via Zoom, by phone, or in-person at home or in a public space. They could be interviewed together or separately, and have other household members present as they wished. Patients and caregivers were interviewed by LF *(female, Samoan, clinical and research experience, Bachelor of Medicine and Bachelor of Surgery)* and were given the option of being interviewed in English or Samoan language. These interviews followed a Pacific conversational model of enquiry called 'talanoa'. Talanoa is ancient practice involving multi-level and multi-layered critical discussions and free conversations without a rigid framework. Talanoa enables social conversations which may lead to critical discussion, producing knowledge and possibilities for addressing issues that affect Pacific people [27]. LF is experienced in conducting talanoa conversations. All other participants were interviewed in English online, one-on-one, by JO *(female, New Zealand-European, qualitative research experience, PhD in public health)*.

## Focus group

We re-contacted eligible interviewed participants and invited them to participate in an online youth focus group held over Zoom using a semi-structured interview guide (S4 Appendix) with the audio recorded. LF introduced topics and chaired the session. JO observed and took notes.

Interview and focus group participants were offered a gift card for their contribution.

## Transcription

Audio files were transcribed with identifiers removed (transcripts were re-identifiable to the interviewers only). Participants were able to review and edit their interview transcript on request; three did so. Where participants were interviewed in Samoan, the interviewer produced a translated English-language transcript.

## Analysis

We performed an initial inductive descriptive thematic analysis. After reviewing the transcripts, data were categorised into codes and sub-codes, then codes were grouped thematically using NVivo (v12). Two researchers (JO, LF) used a virtual whiteboard (miro.com) to identify common and unique themes and sub-themes, then refined and named themes. Themes were refined with advice from co-investigators and the PRG. Final themes were organised deductively within the TYMS Pacific Youth Wellbeing Framework (Fig 1).

Verbatim quotes are provided to support thematic analysis. Some quotes were modified slightly to assist readability.

## Results

### Description of participants

Twenty-seven participants were interviewed: nine patients and nine caregivers (all Samoan), a GP, three cardiologists, a paediatrician, a health policy advisor, an epidemiologist and two stakeholders with experience managing interstate RHD control programs. Seven caregivers were interviewed during the same interview session as the patient participant; one patient was interviewed but their caregiver did not participate; and one patient was interviewed at the same time as their two caregivers. The median patient participant age was 13 years; range 10–22 years. Three patient participants were female and six were male. Three patients and four caregivers were interviewed in-person, all others were interviewed online via Zoom. Of the nine patient participants, six had ARF and three others had required cardiac surgery for RHD. All patient participants reported receiving regular BPG injections except one who had switched to using oral antibiotics. All patient participants granted the researchers permission to access their RCH medical record. The researcher was unable to contact two of ten caregivers who consented to being contacted. Sixteen of 17 GPs did not respond to the invitation, nor did three other stakeholders.

Interviews ranged from 24 to 71 minutes. The average interview duration was 48 minutes for patient-caregiver sessions, and 41 minutes for others. No patient/caregivers were previously known to LF. Six stakeholder participants were known to JO through their professional roles. Three caregivers were interviewed in Samoan. All other interviews were in English. Three interviewed patients were eligible and were invited to take part in the youth focus group, two of whom attended (both were male). The online focus group ran for 52 minutes.

### Key themes

**Wanting to know more about rheumatic heart disease (corresponding TYMS Framework domains: cognitive, and social relationships).** Patients and caregivers did not describe themselves as RHD experts, saying they did not feel they knew much about ARF/RHD, but expressed a strong desire to learn more. Despite this, some went on to provide accurate descriptions of the causal pathway. The need for secondary prophylaxis was unanimously

accepted, with most patients and caregivers linking regular secondary prophylaxis with preventing cardiac damage, and/or staying well. Participants shared questions about their recommended secondary prophylaxis duration and how ARF/RHD affected their health. Young adults, in particular, wanted to know more about their condition and ways to be healthy. A desire for information to inform healthy eating and exercise schedules was expressed. Some patients discussed their perception that ARF had weakened them and wanted to recover their strength.

*'It [ARF] makes you weak, makes you change who you are. It's like a very sad thing to have because it doesn't make you reach your full potential.' Patient #1*

Patients often said they did not like to dwell on ARF/RHD because they felt sad thinking about it and had unpleasant memories of being very unwell. Several caregivers had prior knowledge of ARF from a diagnosis in their family. Several others described being surprised to learn they had affected family members after their child was diagnosed, and wished that they had known about interventions to prevent ARF before their child became unwell. Participants discussed sharing little, if anything, about having ARF/RHD with anyone outside of their family and healthcare providers, feeling that others would not be able to relate. Many said they wanted to meet others who had been affected by ARF/RHD to learn from them and seek reassurance. Several caregivers enquired about a support group.

*'. . .just going back to the community support groups. . . . it's a bit hard to talk about it because she [patient] doesn't understand [ARF]. Maybe if there are other people on this journey that did understand. . . [she could] see what they're doing to help make themselves better.' Caregiver #1*

**Improve clinicians' understanding of RHD in Victoria (corresponding TYMS Framework domains: physical health and mental health).** A strong desire to support clinicians to increase their awareness of ARF/RHD and which groups are at-risk in Victoria was expressed. While some participants had been diagnosed correctly and treated at their initial presentation, others reported the ARF/RHD diagnosis being missed repeatedly by GPs and hospital emergency department staff, while experiencing distressing symptoms. These experiences were traumatic for patients and families. Affected participants shared their outrage and frustration, with caregivers remembering feeling distressed at having to insist that their child be thoroughly and repeatedly assessed by healthcare providers. Many said they felt let down by medical staff.

*". . .Why am I even listening to this person [ED doctor] saying he's okay to go to school? . . .As a mother, my mind was already distraught, it was already shocked, it was already overwhelmed with what I was having to go through. . . I couldn't breathe properly. They [my GP] put it down to anxiety from trauma [from a repeatedly missed RHD diagnosis]. . ." Caregiver #2*

In some cases, the child's condition had deteriorated by the time the correct diagnosis was made, leading to a loss of trust in healthcare providers.

*"When I complained about the GP, he said, "I'm really sorry about your case, but it's just doctors over here are not familiar with rheumatic disease," and that's pretty much where the problem is. . . .if they'd found it out early, his heart wouldn't be [damaged]. . . and if we kept listening to that doctor, we would've lost him. They said it was almost too late." Caregiver #3*

After the ARF/RHD diagnosis, caregivers sometimes described their frustration from challenging GPs' patient management, and insisting the patient/their siblings undergo a throat swab when presenting to primary care with pharyngitis, and receive antibiotics if *S. pyogenes* was detected. Two participants described separate occasions where a highly adherent patient was discovered to have been receiving short-acting benzylpenicillin injections in error over multiple years, when they should have been receiving BPG.

Other stakeholders, including clinicians, agreed there was a need for greater awareness of ARF/RHD among GPs and ED staff in areas of Victoria where high-risk populations are clustered. Supporting clinicians to understand key topics was emphasised, in particular ARF/RHD symptoms, primary and secondary prevention, and which children are at highest risk.

**Pacific patients need more support to receive care (corresponding TYMS Framework domains: environment, resources).**   Most participants were aware that Pacific people were at higher risk of ARF but lacked confidence in the Victorian healthcare system's capacity to detect cases (except at RCH), and facilitate ongoing case management. A lack of coordination between treating physicians and restrictions around sharing medical records were identified as barriers to patients transitioning from paediatric to adult care systems without being lost to follow up, and as barriers to primary and tertiary healthcare providers liaising effectively. Several interviewed clinicians suggested that ensuring patients' GPs were highly engaged in their treatment. They also postulated that continuing to involve family members in the patient's care as they transitioned into the adult care system could reduce loss to follow up, and improve the quality of care. Some patients and caregivers described feeling burdened by having to explain their medical history repeatedly to clinicians.

*"I was overwhelmed starting to think all this took so long because I wasn't recalling everything. I ended up documenting in a book what actually had happened, just so it was easier if they had asked me again, what were signs that were happening, symptoms." Caregiver #2*

Participants who had immigrated to Australia discussed their confusion around how to obtain secondary prophylaxis in Victoria and obtain medical treatment more generally. They shared concerns around the cost of care, particularly expenses associated with purchasing BPG and hospital parking. Participants reported paying differing amounts for BPG. While all said they would prioritise finding the money, many were concerned the cost of BPG would increase.

"There is not much support to help us. Even finding ways to make ends meet for our children, maybe there should be some sort of support like vouchers for families." Caregiver #4

One young adult patient discussed a belief that they would need to find an after-school job to afford their secondary prophylaxis.

"When found out the medication costs money, I was like, "Oh, damn! That sucks." . . .I'm going to have to get a job, a form of transport too. I guess it's going to be hard to do that." *Patient #4*

**Patients will always have family support (corresponding TYMS Framework domains: family and culture).**   Patients identified family as their primary source of support and security. Caregivers described their extended family providing support to manage their child's health. Despite often managing busy households with many children, caregivers were highly invested in their child's diagnosis and treatment, and all identified getting secondary prophylaxis on time as a top priority. Participants were confident about their ability to stay up-to-date

with BPG injections. Most reported never missing an injection, although a few caregivers shared feelings of guilt at having missed a single injection.

*"There was one missed injection during [COVID-19] lockdown. . . .I completely forgot and I felt so bad about it. We have never missed an injection since."Caregiver #5*

Older patients and caregivers discussed patients taking on more responsibility for their treatment as they approached adulthood. Such responsibilities included organising and attending appointments alone, paying for BPG and having conversations with clinicians. Despite this, caregivers said they would always be there to support their child to receive the treatment they need.

*"For us, with our community and being an Islander, that age [18 years] means nothing. That support will always be there. Either me or my sister will continue . . .to take him to his appointments [to have BPG injected]." Caregiver #6*

Patients were largely confident about taking on more responsibility, knowing that they could always look to their family for assistance. Most patients described feeling apprehensive visiting hospitals, especially by themselves, but mentioned that they liked their hospital-based physicians and GP providers. The oldest patient shared how he missed, and is now putting off, his first appointment with an adult cardiology specialist, but intends to re-book because his parents would want him to attend.

*". . . it's my responsibility to take care of myself and drive myself there. I sort of have flashbacks of when I was in the hospital and I don't really like that feeling. . . .it doesn't really motivate me to go. . . .but if I was to tell them [my parents] that I haven't been to the doctor's to get a check-up, they definitely won't be happy. So, I definitely have to call back." Patient #2*

**Inconvenience and injection pain is worth it (corresponding TYMS Framework domains: physical health, mental health).**   Patients and caregivers were strongly motivated to receive all the recommended BPG injections, but injection pain was the first thing they all wanted to change about treatment delivery. Severe injection pain during, and following, BPG administration was described. Patients expressed confidence in their ability to tolerate the injection. Caregivers shared feeling distressed watching their child receive the injection. Patients described a range of techniques to get through injection administration, such as distractions, numbing the skin first with aesthetic cream or ice, and having something to look forward to afterwards. Most said they wanted to rest afterwards and avoid putting weight on the injected area due to soreness.

*"After, I go eat at a fast food restaurant. [laughs] That's my treat. I put a pillow on it [injection site], and I sleep on the side where it doesn't hurt." Patient #3*

Some participants described lignocaine being added to BPG, saying this helped a lot but the area still hurts afterwards. A range of injection sites were described (including the upper arm), with variable administration speeds and inconsistent bruising. One participant had switched to twice daily oral antibiotics due to a needle phobia, and discussed difficulty remembering to take the pills. A strong desire for more convenient and less painful secondary prophylaxis was expressed, including by having injections at home, or requiring secondary prophylaxis less frequently, or over a shorter timeframe.

Patients and caregivers described their motivations to receive BPG, saying they wanted to avoid the unpleasant symptoms which had triggered the ARF/RHD hospitalisation and hoping to avoid cardiac damage. Caregivers of ARF patients often expressed relief their child was receiving BPG and hope that they would avoid lasting heart damage.

*"I get monthly injections now. I don't mind getting them. They're sort of just there. If I don't want to end up like I did in 2018 [undergoing heart surgery], then I might as well just keep taking the injection." Patient #4*

Inconvenience having to organise and remember clinic appointments for BPG was widely discussed, although treating clinics regularly sent reminders.

*"Yes, its expensive [BPG], but we pay it because it's for our children [two of whom had ARF]. . . .we take turns taking our children to their appointments. We make sure one of us is available. We take a day off from work if the appointment times are early in the mornings." Caregiver #1*

Patients and caregivers agreed that an online app with information about RHD and capacity to help them keep track of appointments would be useful, particularly if it could include specialist appointments.

## Trusted GPs are treasured (corresponding TYMS Framework domain: relationships)

Close relationships with GPs were enjoyed by some participants. Clinicians who took the time to demonstrate their understanding of the patient's condition and build rapport were highly valued. This was particularly so following a loss of trust in healthcare providers after the ARF/RHD diagnosis had been repeatedly missed at first. Caregivers described relief at finding a primary care provider who they felt was competent to treat their child.

*"Thank God we found this doctor. . . .He knows exactly what the kids go through, especially with this type of condition." Caregiver #3*

Caregivers shared their willingness to travel to a non-local clinic to have BPG administered by their preferred provider, and would follow their provider if they moved to a new practice.

*"After [patient] was diagnosed, we changed our GP. . . ..he looks after [child] well, and we trust him. So, that's the reason why we travel all the way there. Caregiver #7*

Even when patients had trusted relationships with healthcare providers, many said they felt too shy to ask them questions. Interviewed clinicians described feeling uncertain of how well their patients understood ARF/RHD, and discussed the importance of building rapport with families within a fast-paced and impersonal medical setting. Older patients discussed wanting to overcome shyness and have two-way conversations with healthcare providers, with caregivers wanting to support this.

*"It was always me having to do the talking . . .but now that he is 18 we got told he would transition to the adult clinic. . . Now we have to really get him to start talking!" Caregiver #2*

## Discussion

A clear need to improve the delivery of clinical and public health services for people living with ARF and RHD in Victoria was apparent. Ensuring high uptake of secondary prophylaxis over many years is challenging and requires an understanding of local factors that affect high-risk groups [28,29]. Key barriers to receiving treatment and monitoring which we identified included financial costs, inconvenience of attending appointments, limited clinical awareness of ARF/RHD, and patients feeling unable to have two-way conversations with healthcare providers. Identification of these barriers has, in turn, enabled us to identify potential solutions and make recommendations to improve the delivery of clinical care (Table 1). Introducing an RHD register to Victoria which is supported by a specialist RHD nurse could help to overcome many of these barriers. A well-resourced register would enable patients to maintain their continuity of care when moving between healthcare providers, and routinely collected data could facilitate future assessments of the effectiveness of BPG delivery [7,30,31]. In addition, a list of clinics that provide BPG could be made publicly available as a patient resource. Removing the costs of purchasing BPG currently borne by patients and their families would enhance the accessibility of secondary prophylaxis. A positive relationship between patients, caregivers and healthcare providers is well-established as a determinant of BPG adherence in other Australian jurisdictions and overseas [24,32,33]. Continuity of patient care could be supported by funding a specialist RHD nurse who is capable of answering clinical questions. Given the relatively small number of patients receiving RHD prophylaxis [10], one nurse could potentially serve the entire Victorian patient population. The nurse may have a dual responsibility to support treating clinicians to provide best practice care. They could organise workshops and disseminate educational modules that increase clinical awareness of ARF and RHD. Increased clinical awareness may reduce the delays in diagnosis which a number of participants identified as contributing to poor health outcomes.

Building trust with a healthcare provider is an important aspect of chronic disease management [29]. Pacific people may tend to avoid seeking Western medical treatment unless they are seriously unwell [19,21]. Experiencing racism, feeling stereotyped and judged in healthcare settings has been identified as a barrier to their health service uptake in Australia [14,20]. Pacific ARF/RHD patients having sensitive and engaged healthcare providers may help to overcome this, and help address feelings of shyness inhibiting two-way clinical conversations. Clinicians involving the family in patient care even once the patient reaches adulthood may be an acceptable way to support ongoing treatment and monitoring [29]. Evidence for this is demonstrated by interviewed caregivers expressing an intention to always support their child to receive treatment, and patients identifying family as their primary support. A high mental load from frequent clinic appointments was discussed by participants. This could be reduced, in part, if hospitals were to co-ordinate multiple outpatient clinics to take place on the same day (e.g. with an infectious disease physician and a cardiologist), and ensure that other relevant health staff (such as nutritionists) were also available.

Beyond acknowledging that local Pacific communities are over-represented in ARF, targeted prevention efforts are needed. Although this study focused on children, many of the issues raised are likely to be relevant to Pacific adults with RHD living in Australia, including those on agricultural work visas in rural/remote areas with limited primary health care and no access to Medicare or PBS-funded medications. Sharing Victorian-specific patient resources with patients moving into the state could help engage them with healthcare and promote secondary prophylaxis adherence. A support group may also be appreciated as a patient resource [32]. Many support groups for people with chronic health conditions exist in Victoria and provide opportunities for members to learn about their condition, share experiences and socialise

**Table 1. Recommendations for improving Victorian health policies, systems and clinical practice for Pacific children and young people living with acute rheumatic fever and rheumatic heart disease.**

*Theme*: *Wanting to know more about rheumatic heart disease*

**Create Victoria-specific patient resources**
- Ensure resources are culturally appropriate for, and tailored to, the Victorian patient population.
- Help people understand ARF, RHD and case management in Victoria.
- Make resources easily obtainable for patients coming into Victoria.
- Provide Victorian patients with links to the Heart Foundation's Champions4Change program.

**Create and maintain a support group for patients and families**
- This could be hosted through a not-for-profit stakeholder organisation and may be independent from clinicians.
- The support group does not necessarily need to be Victoria-specific.

*Theme*: *Pacific patients need more support to receive care*

**Implement a Victorian patient register to facilitate best practice management**
- Permit rapid record sharing between healthcare providers.
- Enhance patients' continuity of care.
- Enable a list of clinics that provide BPG to be made publicly available.
- Include cardiac status on the register which clinicians may view, and edit.
- Automate data collection wherever practical.

**Introduce a specialist Victorian RHD nurse**
- This nurse could liaise with treating clinicians and support them to provide best-practice patient care, including by reducing injection pain as much as possible.
- By building close relationships with patients and families, the nurse could help answer clinical questions, and facilitate the patient journey, including the transition to adult care.

**Support GPs and ED staff in areas where Pacific and other high-risk groups are concentrated to increase their awareness of ARF, RHD and risk factors**
- Develop continuing education modules.
- Awareness raising could be supported by a RHD nurse and not-for-profit stakeholders.

**Enhance RHD prevention measures**
- Promote ARF awareness and prevention in at-risk Victorian communities through culturally appropriate community-led interventions.
- Support migrants to obtain affordable healthcare.
- Encourage community members to talk about ARF.
- Ensure Pacific leadership and representation occurs across all aspects of Victorian RHD control.

*Theme*: *Injection pain is worth it*

**Make getting BPG simple and free of charge**
- Reduce confusion around costs, investigate and remove costs to consumers.
- Ideally enable flexible BPG delivery at home.
- Support development of technologies that make secondary prophylaxis more convenient and less painful.
- Support clinicians to provide BPG injections using techniques that minimise pain.

*Theme*: *Trusted GPs are treasured*

**Maintain close clinical involvement in patient care**
- GPs should be involved in patient care—even if not administering BPG personally.
- With a specialist RHD nurse, GPs should maintain links with family who may help re-engage youth in healthcare if necessary.
- Clinicians can link patients to other health and support services, e.g. if moving to a new area–ideally through a register.
- Increase the frequency that adolescent patients are seen by managing GPs and specialists when they are transitioning to the adult care system.
- Implement multidisciplinary hospital clinics which allow patients to see their cardiologist, physician and other relevant staff during a single trip.

[34]. For example, Heartbeat Victoria Council Inc. is a not-for-profit organisation which hosts a number of regional support groups for people with heart conditions [35].

Selection bias in our study is likely because people who were engaged in healthcare attended the RCH clinic and were willing to be interviewed. Clinic appointments provide an opportunity to link patients with prophylaxis and educate them about their condition. Perspectives of people who were unwilling or unable to attend their clinic appointment were missed by our study. This may have accounted for the high BPG uptake reported, as may response bias when speaking with the interviewer. Furthermore, people who were contactable by both the

screening clinician and the researcher may have had stable living situations, which may relate to experiencing less socioeconomic disadvantage than others. Despite this, concerns around the cost of treatment and missed BPG doses were discussed, indicating that participants were comfortable sharing with the interviewer. BPG uptake data was not available in RCH records and therefore we are unable to verify the reported adherence. Employing an ethnically-matched interviewer who was fluent in Samoan and possessed a deep understanding of Samoan cultural practices, including talanoa, was a key strength of our study. Such strengths could be translated to a RHD control program, should a specialist RHD nurse who possessed these qualities be employed. Our interviewer's experience working in healthcare in Samoa likely assisted in putting participants at ease when discussing personal and emotionally difficult topics. Employing ethnically-matched interviewers is thought help to create a culturally safe interview environment which may enhance the experience for the participant and the quality of information they provide [36]. It is unclear whether this occurred in our study.

While people of all Pacific ethnicities were able to participate, interviewed patients and caregivers all had Samoan backgrounds. The study findings may therefore not be extrapolatable to other patient groups. Most domains of the TYMS Framework were reflected in our interviews, indicating this framework was appropriate for this study. Participants described their family as their support base, as the model indicates, and cultural expectations around youth gaining responsibility and independence were discussed. While the TMYS domain includes spirituality as a pillar of youth wellbeing, this aspect was not generally mentioned by participants—beyond expressing gratitude that their RHD experience had not been any worse.

There was a very poor response rate among invited GPs (17/18 did not respond), despite three invitations to participate. This likely reflects the busy Victorian primary healthcare setting, and possibly a lack of engagement with RHD. A study of young Pacific mothers in Victoria noted that contemporary healthcare models made it difficult for them to develop trusted relationships with healthcare providers [20]. These participants suggested that models of care that were effective in New Zealand should be replicated in Australia. Notably this included providing Pacific health workers, outreach services and greater service integration [20,23]. These findings also hold relevance to Pacific families with ARF/RHD in Victoria, who frequently mentioned preferring to receive BPG at home [24,32].

Recommendations for changes to Victorian health policies, systems and practices are made below based on our findings (Table 1).

## Conclusion

Unlike in northern Australia, around half of the patients presenting with ARF in Victoria have Pacific ethnic backgrounds. There are clear opportunities to improve the delivery of clinical care and public health management for people living with ARF/RHD in Victoria, particularly by increasing Pacific leadership and stakeholder representation. Introducing a patient register and a specialist RHD nurse to support best practice patient management may help to improve the continuity of care. Supporting clinicians who work with at-risk populations to increase their awareness of ARF/RHD should also be a key focus, as well as creating and disseminating patient resources that are specific to the unique Victorian patient population. There is potential to help address the marginalisation of Victorian Pacific communities by putting patients' and their families' needs at the forefront of new policies to improve health outcomes.

## Supporting information

**S1 Appendix. Interview guide–patients and caregivers.**
(DOCX)

**S2 Appendix. Interview guide–clinicians.**
(DOCX)

**S3 Appendix. Interview guide–other stakeholders.**
(DOCX)

**S4 Appendix. Focus group guide.**
(DOCX)

## Acknowledgments

The researchers acknowledge the participants and project reference group members who gave their time to this research. We also acknowledge Dr Jessica Kaufman (MCRI) who critically reviewed the study protocol and advised on the study design.

## Author Contributions

**Conceptualization:** Daniel Engelman, Andrew C. Steer.

**Data curation:** Jane Oliver, Loudeen Fualautoalasi-Lam, Andrew C. Steer.

**Formal analysis:** Jane Oliver, Loudeen Fualautoalasi-Lam, Angeline Ferdinand, Ramona Tiatia, Bryn Jones.

**Funding acquisition:** Jane Oliver.

**Investigation:** Jane Oliver, Loudeen Fualautoalasi-Lam, Andrew C. Steer.

**Methodology:** Jane Oliver, Loudeen Fualautoalasi-Lam, Angeline Ferdinand, Ramona Tiatia, Bryn Jones, Daniel Engelman, Katherine B. Gibney, Andrew C. Steer.

**Project administration:** Jane Oliver, Loudeen Fualautoalasi-Lam.

**Resources:** Jane Oliver, Angeline Ferdinand, Andrew C. Steer.

**Software:** Jane Oliver.

**Supervision:** Jane Oliver, Ramona Tiatia, Katherine B. Gibney, Andrew C. Steer.

**Writing – original draft:** Jane Oliver.

**Writing – review & editing:** Loudeen Fualautoalasi-Lam, Angeline Ferdinand, Ramona Tiatia, Bryn Jones, Daniel Engelman, Katherine B. Gibney, Andrew C. Steer.

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
