## [Decision Letter · Decision Letter 0]

21 Jun 2024

Dear Dr. Oliver,

Thank you very much for submitting your manuscript "Living with rheumatic fever and rheumatic heart disease in Victoria, Australia: a qualitative study" for consideration at PLOS Neglected Tropical Diseases. As with all papers reviewed by the journal, your manuscript was reviewed by members of the editorial board and by several independent reviewers. The reviewers appreciated the attention to an important topic. Based on the reviews, we are likely to accept this manuscript for publication, providing that you modify the manuscript according to the review recommendations. 

Sincerely,

Husain Poonawala

Academic Editor

Victoria Brookes

Section Editor

Reviewer's Responses to Questions

**Key Review Criteria Required for Acceptance?**

**Methods**

-Are the objectives of the study clearly articulated with a clear testable hypothesis stated?

-Is the study design appropriate to address the stated objectives?

-Is the population clearly described and appropriate for the hypothesis being tested?

-Is the sample size sufficient to ensure adequate power to address the hypothesis being tested?

-Were correct statistical analysis used to support conclusions?

-Are there concerns about ethical or regulatory requirements being met?

Reviewer #1: Well-designed qualitative study.

**Results**

-Does the analysis presented match the analysis plan?

-Are the results clearly and completely presented?

-Are the figures (Tables, Images) of sufficient quality for clarity?

Reviewer #1: The results follow logically from the methods and are clearly expressed.

Three of the four interview guides include questions about how people would respond to different ways of administering BPG but there was nothing about this in the results. Will these results be the subject of a separate paper?

**Conclusions**

-Are the conclusions supported by the data presented?

-Are the limitations of analysis clearly described?

-Do the authors discuss how these data can be helpful to advance our understanding of the topic under study?

-Is public health relevance addressed?

Reviewer #1: The authors have stated the limitations of their study and situated their conclusions within these limitations. 

Another bias is that the recruitment strategy is likely to have biased towards over-selection of metropolitan patients. Based on the notification data available since July 2023, what proportion of patients live outside the metropolitan area and how are their experiences likely to differ from those of study participants?

Given that "All patient participants granted the researchers permission to access their RCH medical record" was any attempt made to triangulate self- and parent- reported adherence with BPG secondary prophylaxis against BPG injections documented in the medical record? There might be social desirability bias in reporting high adherence and this part of the results is a bit discordant with the barriers reported.

**Editorial and Data Presentation Modifications?**

Reviewer #1: Typo in line 434 - " ethically-matched" should probably be replaced by " ethnically-matched"

**Summary and General Comments**

Reviewer #1: Overall a well-designed and implemented qualitative study which provides robust evidence to advocate for a state-based RF/RHD register and prioritisation of improving RF/RHD prevention and treatment services to Pasifika people. Although this study focused on children, many of the issues raised are likely to be as, if not more, relevant for adult Pasifika people with RHD living in Australia as many are on agricultural work visas, in rural/remote areas with limited primary health care and do not have access to Medicare or PBS-funded medications.

PLOS authors have the option to publish the peer review history of their article (what does this mean?). If published, this will include your full peer review and any attached files.

Reviewer #1: No

Figure Files:

Data Requirements:

Reproducibility:

References

---

## [Decision Letter · Decision Letter 1]

8 Aug 2024

Dear Dr. Oliver,

We are pleased to inform you that your manuscript 'Living with rheumatic fever and rheumatic heart disease in Victoria, Australia: a qualitative study' has been provisionally accepted for publication in PLOS Neglected Tropical Diseases.

Best regards,

Husain Poonawala

Academic Editor

Victoria Brookes

Section Editor

Reviewer's Responses to Questions

**Key Review Criteria Required for Acceptance?**

**Methods**

-Are the objectives of the study clearly articulated with a clear testable hypothesis stated?

-Is the study design appropriate to address the stated objectives?

-Is the population clearly described and appropriate for the hypothesis being tested?

-Is the sample size sufficient to ensure adequate power to address the hypothesis being tested?

-Were correct statistical analysis used to support conclusions?

-Are there concerns about ethical or regulatory requirements being met?

Reviewer #1: (No Response)

Reviewer #2: (No Response)

**Results**

-Does the analysis presented match the analysis plan?

-Are the results clearly and completely presented?

-Are the figures (Tables, Images) of sufficient quality for clarity?

Reviewer #1: (No Response)

Reviewer #2: (No Response)

**Conclusions**

-Are the conclusions supported by the data presented?

-Are the limitations of analysis clearly described?

-Do the authors discuss how these data can be helpful to advance our understanding of the topic under study?

-Is public health relevance addressed?

Reviewer #1: (No Response)

Reviewer #2: (No Response)

**Editorial and Data Presentation Modifications?**

Reviewer #1: Typos in lines 50, 148, 617 and 632 identified by spell check in the Word version with track changes.

Reviewer #2: (No Response)

**Summary and General Comments**

Reviewer #1: Thank you for revising the paper in response to the comments.

Reviewer #2: I am happy with he changes made by the reviewers and recommend the manuscript be published

PLOS authors have the option to publish the peer review history of their article (what does this mean?). If published, this will include your full peer review and any attached files.

Reviewer #1: No

Reviewer #2: **Yes: **Emma Haynes

---

## [Editor Report · Acceptance letter]

26 Aug 2024

Dear Dr. Oliver,

We are delighted to inform you that your manuscript, "Living with rheumatic fever and rheumatic heart disease in Victoria, Australia: a qualitative study," has been formally accepted for publication in PLOS Neglected Tropical Diseases.

Best regards,

Shaden Kamhawi

co-Editor-in-Chief

Paul Brindley

co-Editor-in-Chief
